# Feeding and Management of Horses with and without Free Faecal Liquid: A Case–Control Study

**DOI:** 10.3390/ani11092552

**Published:** 2021-08-30

**Authors:** Katrin M. Lindroth, Jan-Erik Lindberg, Astrid Johansen, Cecilia E. Müller

**Affiliations:** 1Department of Animal Nutrition and Management, Swedish University of Agricultural Sciences, P.O. Box 7024, 750 07 Uppsala, Sweden; Jan.Erik.Lindberg@slu.se (J.-E.L.); cecilia.muller@slu.se (C.E.M.); 2NIBIO, Norwegian Institute for Bioeconomy Research, P.O. Box 115, NO1431 Ås, Norway; astrid.johansen@nlr.no

**Keywords:** equine, free faecal water, nutrition, watery faeces, wrapped forage

## Abstract

**Simple Summary:**

Horses with free faecal liquid defecate in one solid and one liquid phase, and the liquid phase can be a concern for the horse owner and veterinarians. The causes of free faecal liquid are unknown, but previous studies have indicated that feed ration composition may play an important role in the occurrence of the condition. A study comparing feed rations, feeding practices and management factors for horses with and without free faecal liquid was performed. Horses without free faecal liquid were reported to have a lower daily intake of starch and sugar and a higher daily intake of protein and fibre compared to horses with free faecal liquid. Horses with and without free faecal liquid were fed similar amounts of wrapped forages and were subject to the same management practices. The reported differences may be of importance for the condition, but further studies are required to establish if its occurrence is due to specific feeding regimens.

**Abstract:**

Free faecal liquid (FFL) in horses is characterised by the excretion of faeces in two phases (one solid and one liquid), which may cause dermatitis on the hindlegs. The causes of FFL are not known. Results from previous studies have indicated that feed ration composition and management factors may play important roles in the occurrence of FFL. A case–control study was therefore performed in which data on feed rations, feeding practices and management factors were compared between horses with (case) and without (control) FFL on 50 private farms in Sweden and Norway. The comparisons show that case and control horses were reported to be fed similar average amounts of wrapped forage (*p* = 0.97) and to be subject to similar management practices, but case horses were fed higher proportions of concentrates in their diet (*p* < 0.001) and lower average amounts of straw and lucerne (*p* < 0.05) compared to control horses. Case horses were reported to be fed twice as much concentrate per 100 kg BW and day as control horses and a higher daily intake of starch and water-soluble carbohydrates (*p* < 0.05). Case horses also had a lower daily intake of digestible crude protein and neutral detergent fibre compared to control horses (*p* < 0.05). These differences were small but are of interest for further studies of factors causing FFL.

## 1. Introduction

During the past decade, a condition known as free faecal liquid (FFL) or free faecal water syndrome (FWS) in horses has gained attention [1,2,3]. The condition is characterised by the two-phase separation of the horse faeces, one solid and one liquid phase, which may be voided together or separately [4]. The causes of the condition have not been identified, but several factors related to specific feeds, feeding practices and management regimes have been suggested. Among feed-related factors, the use of wrapped forages or high amounts of alfalfa hay in the equine diet has been proposed as a possible cause of FFL [1]. In a case report on one horse with FFL, factors such as increasing the number of feedings per day, change in forage type (e.g., harvest number, botanical composition) and changes in management routines were part of resolving the condition [3]. However, no systematic studies of these factors and their possible association with the presence of FFL are available.

The occurrence of FFL was previously reported in horses fed hay ad libitum along with high grain content (>4 kg every 12 h) in the feed ration, but it was not observed in the same horses fed only hay ad libitum [5]. In a survey of 339 horses with FFL and their feeding and management practices, it was found that changing from one batch of haylage (wrapped grass forage with 500–840 g dry matter (DM)/kg) to another batch, to grass hay (≥840 g DM/kg) or to the provision of pasture grass resulted in a reduction in or elimination of FFL symptoms in 17, 58 and 46% of the horses, respectively [4]. This indicates that feed ration composition and forage type may play important roles in the occurrence of FFL. The aim of the present study was to compare feeding practices, feed rations and management factors between horses with (case) and without (control) FFL in order to identify factors associated with this condition. The hypothesis was that horses with FFL were fed differently compared to horses without FFL, with a lower amount of forage and a higher amount of concentrate in the daily intake of FFL horses.

## 2. Materials and Methods

### 2.1. Horses

A case–control study was performed on horses from 50 privately owned farms (30 in Sweden and 20 in Norway), with one case and one control horse on each farm. A case horse was defined as a horse showing FFL (faeces with one solid and one liquid phase), whereas a control horse was defined as a horse not showing signs of FFL. The horses in each pair were fed the same wrapped forage and kept in the same stable or loose housing system and in the same or adjacent paddocks. Other inclusion criteria were that all horses were over two years of age, were not subjected to any recent changes in feed or farm, did not show any signs of ongoing infection (no pyrexia), had received no medical treatment during the preceding six months and showed no clinical signs of any gastrointestinal tract disturbances during the preceding six months. All horse owners provided written informed consent before entering the study. All data were handled according to the General Data Protection Regulation Act in the European Union. No ethics approval of animal experiments was required for this study according to EU or national legislations.

### 2.2. Sampling and Analysis of Wrapped Forages

The forage on each farm was sampled on three occasions: October/November 2016, January/February 2017 and March/April 2017. Prior to each sampling period, horse owners received a kit with sampling materials and detailed, illustrated instructions on how the samples should be collected and handled. Horse owners were asked to collect approximately 250 g of forage by grab sampling from the forage that the horses were fed. All samples were sent by post to the Department of Animal Nutrition and Management, Swedish University of Agricultural Sciences (Uppsala, Sweden), for chemical composition analysis. If time from sampling to sample arrival at the department exceeded four days, the samples were discarded. The horse owner was then asked to provide new samples within the timeframe of that sampling period.

### 2.3. Chemical Analysis of Forage

Forage samples were analysed for the contents of dry matter (DM), acid detergent fibre (ADF), neutral detergent fibre (NDF), crude protein (CP), lignin, in vitro digestible organic matter, ash and macro minerals (Ca, P, Mg, Na, K, S). From each sample, 50 g of forage was mixed with equal amounts of water and pressed using a potato press in order to extract juice. From the juice extracted from the forage, the concentrations of volatile fatty acids (VFA), lactic acid, 2,3-butanediol, ethanol and ammonia N and pH were determined. The contents of glucose, sucrose, fructose and fructans were analysed, and the total water-soluble carbohydrates (WSC) were calculated as the sum of these compounds. All analyses and calculations of digestible CP (dCP) content were performed with methods described by Müller et al. [6]. Metabolisable energy for horses (MEh) was calculated from ME for ruminants (MEr), which was estimated from in vitro digestible organic matter content [6].

### 2.4. Data Collection

Basic information about the horses, such as breed, gender, age, coat colour, body condition score (BCS) according to the system of Carrol and Huntington [7], body weight (BW, estimated or weighed), training discipline/intensity, type and amount of feed, feeding practices and horse management, was obtained through an online survey created using the tool Netigate (Stockholm, Sweden) and distributed to the participants. Participants with case horses were also asked to provide information on whether any changes in feeding had been tested and, if so, the type and outcome of these changes. The survey was made available in both the Swedish and Norwegian languages and is provided (translated into English) in Appendix A. Information on the nutritive value of feed other than the wrapped forages (such as concentrates, lucerne chaff/pellets and supplemental feeds) was obtained from the relevant feed companies. Information on the nutritive value of hay (if used) was obtained from analytical reports provided by the horse owners. Data on the chemical composition of straw and grains were obtained from national feed tables [8].

### 2.5. Data Treatment

Over 20 different horse breeds were represented in the dataset. Therefore, the data were grouped into four breed types: warmblood-type horses (European warmblood, Lusitano, Quarter, Standardbred and crossbred warmblood-type horses), cold blood-type horses (cold-blood trotter, crossbred horses of cold-blood type, Dølehorse, Friesian horse and Norwegian fjord horse), hot-blood horses (Arabian and Thoroughbred) and native horse and pony breeds (Icelandic, Lyngshorse, Welsh cob, Welsh pony and crossbred ponies).

The different types of concentrates fed to a particular horse were combined to create one variable, “concentrates”, in the calculations of total amounts of different feeds and proportions of concentrates in the daily feed ration. However, individual nutritive values for each concentrate feed were used to calculate the daily intake of different nutrients and of MEh. Total feed ration composition was calculated for each horse and comprised components that could be determined for all roughages, concentrate feeds and mineral feeds. It was therefore limited to the amount of MEh, dCP, NDF, starch, crude fat, WSC and macro minerals (Ca, P, Mg, Na and K) provided on a daily basis. Horses reported to be fed any feed ad libitum were not included in the calculations of total feed ration composition, daily intake of nutrients and MEh or proportion (% on DM basis) of concentrates in the diet.

### 2.6. Calculations and Statistical Analysis

For statistical analysis, SAS version 9.4 for Windows (Statistical Analysis System Institute Inc., Cary, NC, USA) was used. The data analysed included basic data on horse characteristics, horse management, feeding practices and type and amount of feeds, which were compared between case and control horses by calculating the frequency and using a Chi2 test. For the chemical composition of wrapped forages, the minimum, maximum, median, average and standard deviation were calculated. Analyte values for VFA and lactate below lower detection limits were transformed to half the lower detection limit. The reported estimated BW and the registered daily amount of feeds (in grams or kilograms, depending on feed type) for individual horses were used to calculate the total daily intake of specific nutritional components per 100 kg BW and day.

The daily intake of specific feed components and different feeds was compared between case and control horses using a generalised linear mixed model (GLMM) procedure, with farm (id) included as a random effect:

Y*ij* = μ + (case/control)*i* + (id × case/control)*j* + (error)*ij*, where the term “error” is the random residual with mean = 0 and variance σ^2^.

Missing values were treated as such in statistical analyses. Differences were considered significant at *p* < 0.05, while those at 0.05 ≤ *p* < 0.10 were regarded as tendencies.

## 3. Results

### 3.1. Horses

The distributions of breed type, gender, coat colour and body condition score (BCS) were similar for the case and control horses (Appendix A). The average age was 13 years (SD 5.7) for case horses and 10 years (SD 5.3) for control horses (*p* = 0.45). The majority of both case and control horses were used for leisure riding, but performing multiple disciplines was common for both groups (Appendix A). More case horses tended (*p* = 0.07) to be kept as companion animals compared with control horses (Appendix A). The majority of both case and control horses were reported to perform very-low- to low-intensity exercise (Appendix A).

### 3.2. Chemical Composition of Forages

Analysis of the composition of forage samples showed considerable variation in most variables (Table 1). The variables with the largest discrepancies between mean and median values were sucrose, fructans, lactic acid, butyric acid and ethanol (Table 1). The means of these variables were higher than medians, indicating the presence of some samples with a high content of these components. The majority of participating horse owners reported that they did not know the nutritive content of the forage used (62%, *n* = 31 for case horses and 60%, *n* = 30 for control horses) before entering the study.

### 3.3. Types and Amounts of Roughages

All horses in the study were reported to be fed wrapped forages. The majority of horses (88%, *n* = 44) were fed grass haylage, while 12% (*n* = 6) were fed grass silage (Figure 1). In addition to wrapped forage, the diets of some individuals among both case and control horses contained straw, lucerne and grass hay. The proportions of horses fed the different roughages were similar in case and control horses (*p* = 0.75) (Figure 1). Case and control horses were fed similar average amounts of wrapped forages (*p* = 0.97), but case horses were fed a lower average amount of straw (*p* < 0.0001) compared to control horses (Table 2). Case horses tended to be fed a lower average amount of lucerne (*p* = 0.05) compared to control horses (Table 2).

### 3.4. Types and Amounts of Concentrates

The majority of horses were reported to be fed concentrates (72%, *n* = 36 for both case and control horses). About 50 percent of both case and control horses were reported to be fed commercial concentrates, such as muesli and/or pelleted feeds (Figure 2). Other concentrate feeds reported were vegetable oil, soybean meal, brewer’s yeast (*Saccharomyces cerevisiae*), molassed sugar beet pulp, grains (oats, barley) and wheat bran, which were fed to similar proportions of case and control horses (Figure 2). A higher proportion of control horses compared to case horses were reported to be fed brewer’s yeast (*p* = 0.01) (Figure 2). Control horses were reported to be fed half as much concentrate (total amount) per 100 kg BW and day as case horses (*p* = 0.004) (Table 2).

### 3.5. Types and Amounts of Supplement Feeds

Supplement feeds were reported to be fed to 90% (*n* = 45) of case horses and 78% (*n* = 39) of control horses. The proportions of case and control horses fed any of the supplementary feeds were similar (*p* = 0.83) (Figure 3). The most common supplements were mineral feeds, which were fed to the majority of case (88%, *n* = 44) and control (82%, *n* = 41) horses, while 34% (*n* = 17) of both case and control horses were reported to be fed vitamins. Other feed supplements reported were garlic, gut balancers (a group of products that are marketed as a pre- and probiotic to promote a healthy hindgut of the horse) and rose-hip powder (Figure 3). Case horses tended to be fed a lower average amount of mineral feed (grams per 100 kg of BW per day) in their diets compared with control horses (*p* = 0.08) (Table 2).

### 3.6. Feed Rations and Total Daily Intake of Feed Components

The daily intake of ME_h_, crude fat and macro minerals (Ca, P, Mg, Na and K) was similar between case and control horses (*p* > 0.47) (Table 2). Total daily feed intake (kg DM) tended to be higher (*p* = 0.09) in control horses compared to case horses (Table 2). Control horses were fed a lower proportion of concentrates in their diet compared to case horses (*p* < 0.001). Case horses were fed lower daily amounts of dCP (*p* = 0.007) and NDF (*p* < 0.0001) compared to control horses, whereas daily intake of WSC (*p* = 0.02) and starch (*p* = 0.004) was higher in case horses compared to control horses (Table 2). As daily forage intake was similar in case and control horses, the intake of components attributed to the forage, such as individual and total short-chain fatty acids, ethanol and 2,3-butanediol, was also similar.

### 3.7. Feeding Practices

Similar feeding strategies were reported for case and control horses (Table 3). Over half of all case and control horses were fed forage two to three times daily, while approximately 40% of both groups were fed forage four or more times daily (Table 3). No case horses (0%) and two control horses (4%) had ad libitum access to forage (Table 3). About half of both control and case horses were fed roughage with a maximum of eight hours between two feedings (Table 3). About half of the horses in each group were fed forage in their paddocks in a feed rack, tub, hay-net or similar apparatus, while the remaining horses were fed forage on the ground or not fed forage at all in their paddocks (Table 3). Approximately 60% of both case and control horses were fed concentrates one or two times daily, followed by no concentrate feeding at all (28%) and three or four times daily (12%) (Table 3). Automatic waterers were the most common water source in stables or loose housing systems, while frostless automatic waterers or tubs were the most common in paddocks, with no differences between case and control horses (Table 3).

Participants with case horses reported that previous changes in feeding affected the occurrence of FFL in their horse (Table 4). Changing from primary to regrowth harvests (regrowth harvest defined as 2nd, 3rd or 4th harvest, for wrapped forages), from wrapped forage to hay and from wrapped forage to pasture grass was reported to result in a reduction in or elimination of FFL in 34% (*n* = 17), 24% (*n* = 12) and 18% (*n* = 9) of the case horses (Table 4). Moreover, 26% (*n* = 13) of the case horses were reported to show a reduction or elimination of signs of FFL when adding various types of commercial probiotics and prebiotics, psyllium seed, linseed or thiamine (Table 4).

### 3.8. Management Factors

Management strategies were reported to be similar for case and control horses (*p* > 0.30) (Table 5). The majority of all horses were kept in individual boxes at night and outside in paddocks during the daytime. The second most common strategy was to keep horses in loose housing systems 24/7 (Table 5). Around two-thirds of the horses were kept outside in paddocks for more than 8 h per day, and the remaining horses were kept outside for less than 8 h per day (Table 5). All horses were commonly kept in grass and soil paddocks (about one-third of all horses in each paddock type), followed by forest paddocks and sand/gravel paddocks (Table 5). The most common bedding material used in stables and loose housing systems was straw only or a combination of straw and wood shavings (Table 5). The majority of all horses were reported to be kept on pasture during summer, and about two-thirds spent at least eight weeks on pasture grass (Table 5). Pasture types and water sources on pasture varied between farms but were similar for case and control horses, and most horses on pasture had access to salt licks (Table 5). The majority of case horses and half of the control horses were dewormed on a yearly basis, with high faecal egg counts being the determining factor for deworming (Table 5). Around two-thirds of the horses had been dewormed within six months before the survey was performed, and very few horses were reported to not be dewormed at all (Table 5).

## 4. Discussion

### 4.1. Horses

The distributions of breed types, gender, coat colour and body condition score (BCS) were similar for case and control horses. Most of the case horses were reported to have a BCS of 3 (the ideal BCS in the system of Carroll and Huntington [7]), which is in agreement with previous BCSs reported in horses with FFL [1,3]. A higher proportion of case horses compared to control horses tended (*p* = 0.07) to be kept as companion animals. In a previous study, the majority of horses with FFL (62%, *n* = 22) were reported not to be ridden [9]. It is not possible to draw any conclusions on the cause-and-effect relationship between exercise and the presence of FFL, but as fewer horses with FFL than control horses were in any training, it is possible that the presence of FFL is perceived by the horse owner as a hindrance to training.

### 4.2. Feeding Practices and Management Factors

Feeding practices and management factors were similar for horses with and without FFL. Case and control horses in each horse pair were kept on the same farm and subjected to the same general management and feeding practices, with only minor variations in these variables between case and control horses. No specific feeding or management routine was used for only case or only control horses, indicating that the factors included in this study probably do not play an important role in the occurrence of FFL. Other feeding or management factors not investigated in this study may be of importance for FFL. In a previous case study, increasing the number of feedings per day contributed to resolving FFL [3]. The results of the present study could not verify this, but the results may be difficult to compare, as the studies were performed very differently.

### 4.3. Feeding Forages

Feeding horses wrapped forage has previously been suggested as a possible cause of FFL [1]. In the present study, all case and control horses were fed wrapped forages, and all case–control pairs were fed the same forage batch. Therefore, it is unlikely that wrapped forage per se is the general cause of FFL. However, changes in the forage batch or type were reported to result in a reduction in or elimination of FFL in the current study. Similar results were found in a previous Swedish survey of FFL horses [4], where changes in forage batches were reported to result in the elimination of or reduction in FFL. Studies investigating the influence of different forage conservation methods on forage composition and its impact on the equine hindgut have shown similar results for overall digestibility [10]. In addition, the biochemical and microbial compositions in faeces and in the right ventral colon were similar in healthy horses fed hay, haylage and silage produced from the same grass harvest [6,11]. There are, however, many other factors that may differ between different forage, such as plant maturity at harvest, harvest number and botanical composition, among others. Increased plant maturity results in increased NDF content and decreased fibre digestibility in grasses and legumes [12,13]. The harvest number (primary or regrowth harvest) affects, for example, NDF digestibility and content due to a higher leaf-to-stem ratio in regrowth compared to primary harvests [14].

The botanical composition may also influence CP, NDF and ADF content and their digestibility, depending on the proportions of grasses and legumes in the forage, as legumes generally have higher plant cell wall digestibility compared to grasses [15]. Fibre digestibility of forage may be of interest for further studies on FFL in horses, as there is large individual variation in the hindgut and faecal microbiota composition among horses [16,17,18,19] that may result in variation in the degradation of fibre in individual horses. Fibre composition and degradability in the hindgut may also influence the hydrophilic properties of the ingesta [20,21] and its capacity to hold water, which could in turn affect the presence or absence of FFL. If the proportion of fibre with low water-holding capacity in the hindgut is comparatively large, it is possible that it can result in free liquid that manifests as FFL. In future studies, more detailed information about the forage (including fibre content) before and after a change in the diet of an FFL-affected horse should be included in order to further investigate this hypothesis.

### 4.4. Feeding Concentrates

Case horses were fed a higher amount and proportion (per 100 kg BW) of concentrates compared to control horses. The difference in the proportion of concentrates in the total feed ration was small (9.7% in case horses, 9.1% in control horses) but influenced total daily intake of WSC and starch, which was higher for case horses compared to control horses. The inclusion of grain (high in starch) in diets with *ad libitum* hay intake has been reported to cause two-phase separation of faeces in horses [5]. However, the amount of grain in that study was 4.55 kg every 12 h (a total amount of 9.1 kg grain per day), which was much higher than the amount of concentrates reported in the present study (0.1 and 0.2 kg DM concentrate per 100 kg BW and day for control and case horses, respectively). Nevertheless, smaller amounts of concentrates (2.5–5 kg concentrate per day) have been reported to increase the risk of colic in horses [22,23,24], indicating that lower levels of concentrate feeding could also affect normal gastrointestinal function. If the starch in the feed is not degraded in the small intestine, then it will enter the caecum and undergo microbial fermentation, with subsequent rapid production of lactate in the hindgut [25,26,27,28]. Lactate is poorly absorbed in the gastrointestinal tract [29,30], which may result in the accumulation of lactate to a level exceeding the buffering capacity of the hindgut and result in decreased pH and hindgut acidosis. If the pH falls below six, this favours further microbial production of lactic acid. This increased production has been shown to be associated with osmotic diarrhoea [29], which can occur with a build-up of molecules that attract water into the lumen of the colon [20,21,31]. In the present study, some of the case horses were reported to not be fed any concentrates at all, indicating that osmotic diarrhoea due to starch escaping degradation in the small intestine is not the sole explanation for the presence of FFL. This accumulation and build-up of lactate has also been suggested mechanism in horses with chronic diarrhoea associated with WSC content of the forage [Vergnano et al., 2017]. However, as the same type and amount of forage were fed to the case and control horses and that a change in forage resulted in a reduction of FFL the difference in WSC intake is not due to the forages. However, further studies focusing on osmotic diarrhoea (with different causes) as an important factor in FFL are recommended.

Brewer’s yeast was fed to a higher proportion of control horses compared to case horses in the present study, which may be of interest due to the assumed probiotic effects of many yeast products. However, the efficacy and beneficial effects of yeast as probiotics in equine gastrointestinal disease have been poorly studied [32], and no controlled studies on yeast supplements as a remedy for FFL have been performed. In a case study in which *Saccharomyces cerevisiae* was provided at a dosage of 100 g/day to a horse with FFL, faecal balls were reported to return to normal consistency, but faecal water was still present [3].

### 4.5. Daily Intake of Nutrients and Total Feed Ration

In the present study, small differences were observed in the type and amount of feed. This also resulted in small differences in the daily intake of specific components between case and control horses. Case horses were fed a lower average amount of lucerne and straw compared with case horses, resulting in a lower daily intake of NDF. It is known that the intake of fibre-rich feeds generally enables a steadier rate of production and absorption of intestinal water and nutrients compared to the intake of low-fibre feeds [21,30,33]. Control horses were fed a diet higher in dCP compared to case horses, despite being fed lower amounts of concentrates, which may be attributed to the higher quantity of brewer’s yeast, lucerne or other concentrate feeds high in protein. An excessive CP intake with increased availability of nitrogen, together with increased VFA concentration in the hindgut, could result in increased ingesta water content by inducing an osmotic drive [34]. However, as CP intake was lower in case horses than in control horses, FFL does not seem to be associated with the amount of dCP that the horse consumes.

### 4.6. Limitations of the Study

In this study, all data and samples were provided by horse owners. One limitation of the study could be that different participants may have interpreted the instructions and questions differently. The reported feed rations may also have differed from the actual rations fed, depending on whether the responses referred to what the horse was fed or what was actually consumed. Moreover, some horses were fed in their paddocks, sharing their feed with other horses, which could result in discrepancies in the amount of feed reported compared with the amount actually consumed. These potential discrepancies also apply to the reported daily intake of straw. A number of horses were reported to have straw as bedding material and therefore could have ingested more straw than reported.

## 5. Conclusions

Feed ration composition differed between horses with and without FFL, while feeding practices and management factors did not. Case horses were reported to be fed more sugar and starch and less NDF and dCP compared to control horses. These variables are of interest for further studies on the causes of FFL. In several case horses, the signs of FFL were reported to be eliminated or diminished after changes in the forage batch. More detailed studies on forages, such as harvest number, plant maturity at harvest, botanical composition and chemical composition, and their impact on the signs of FFL are of interest to further investigate.

## Figures and Tables

**Figure 1 animals-11-02552-f001:**
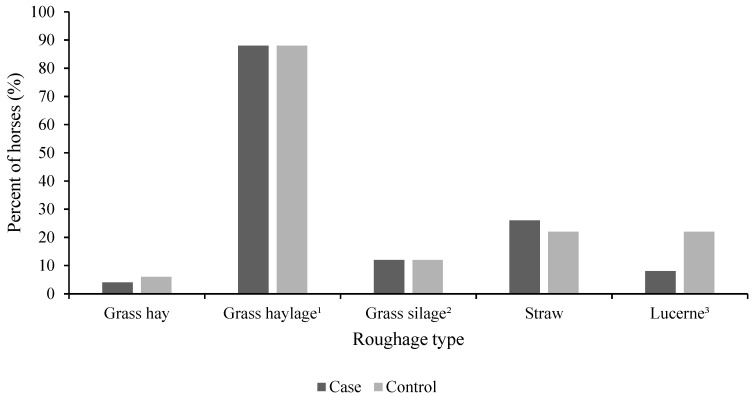
Proportions of horses with (case, *n* = 50) and without (control, *n* = 50) free faecal liquid fed various roughage types. The same horse may have been fed multiple roughages, resulting in a sum exceeding 100%. ^1^ Wrapped forage with ≥50% dry matter. ^2^ Wrapped forage with ≤50% dry matter. ^3^ Includes both pellets and chaff. *p* > 0.05 for all feed types.

**Figure 2 animals-11-02552-f002:**
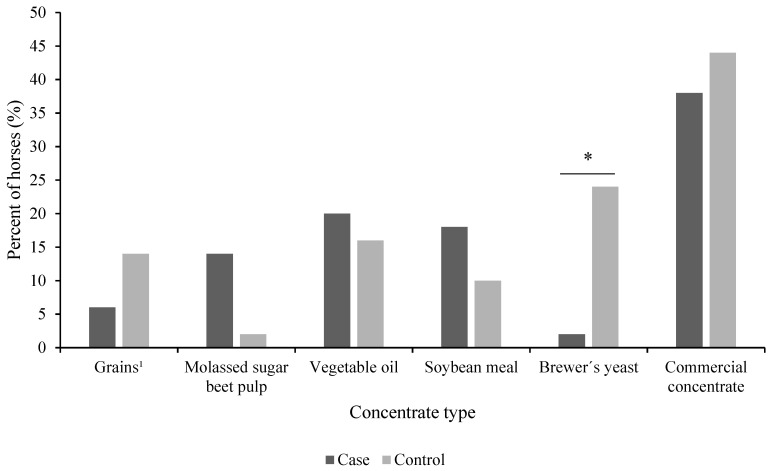
Proportion of horses with (case, *n* = 50) and without (control, *n* = 50) free faecal liquid fed different types of concentrate feeds. Multiple concentrates may have been fed to the same horse, resulting in a sum exceeding 100%. ^1^ Includes oats, barley and wheat bran. A higher proportion of control horses compared to case horses were fed brewer’s yeast (*p* = 0.01). * = *p* < 0.05.

**Figure 3 animals-11-02552-f003:**
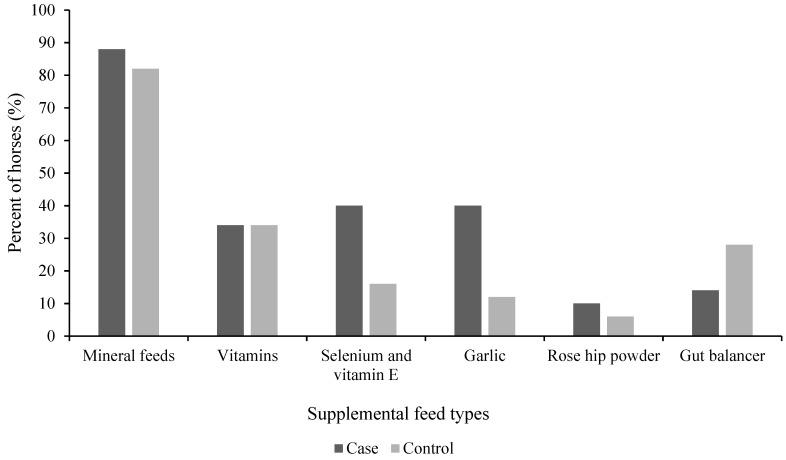
Proportion of horses with (case, *n* = 50) and without (control, *n* = 50) free faecal liquid fed different types of supplement feeds. Multiple supplement feeds may have been fed to the same horse, resulting in a sum exceeding 100%. *p* > 0.05 for all feed types.

**Table 1 animals-11-02552-t001:** Dry matter (DM, g/kg), ammonia N (% of total N), pH, chemical composition, in vitro organic matter (g/kg DM) and metabolisable energy (ME_h,_ MJ/kg DM) in wrapped forages used for the horses in the study. Minimum, median, maximum, mean and standard deviation (SD) values for 50 forage batches with three replicate samples for each forage batch.

Variables	Minimum	Median	Maximum	Mean	SD
Dry matter	179	728	951	692	152.9
Ammonia N	0.003	1.3	5.0	1.6	1.12
pH	3.9	5.5	6.1	5.4	0.46
Ash	29	59	110	60	14.3
Crude protein (CP)	36	88	184	92	28.3
Estimated digestible CP	10	50	139	53	26.4
Neutral detergent fibre	432	607	721	609	49.6
Acid detergent fibre	255	350	931	356	65.5
Lignin	19	34	56	35	7.6
Water-soluble carbohydrates	3	110	230	105	52.7
Glucose	0.1	32.1	132.6	38.1	25.08
Fructose	0.6	38.1	128.6	39.2	33.91
Sucrose	0.2	4.9	113.7	12.1	16.88
Fructans	0.3	12.4	113.7	21.0	22.70
Calcium	1.0	3.5	12.6	3.8	1.64
Phosphorus	1.1	2.1	3.5	2.1	0.48
Magnesium	0.5	1.3	4.0	1.4	0.59
Potassium	5.0	16.7	34.0	17.1	5.03
Sodium	0.05	0.1	2.4	0.3	0.44
Sulphur	0.6	1.5	2.6	1.5	0.41
Lactic acid	0.5	3.5	26.1	5.6	5.08
Acetic acid	0.1	1.1	8.2	1.7	1.83
Propionic acid	0.1	0.2	2.7	0.3	0.35
Butyric acid	0.1	0.1	24.5	2.3	4.84
Formic acid	0.1	0.1	4.3	0.5	0.74
Ethanol	0.1	1.8	30.2	3.6	4.77
2,3-Butandiol	0.1	0.1	9.1	0.7	1.32
Volatile fatty acids	0.4	1.5	39.7	4.8	2.05
Short-chain fatty acids	0.9	5.0	65.8	10.4	6.50
In vitro digestible OM	532	735	867	729	64.4
Estimated ME_h_ ^1^	5.9	9.3	11.4	9.2	1.06

^1^ Estimated ME_h_, estimated metabolisable energy for horses; MJ = mega-joule.

**Table 2 animals-11-02552-t002:** Minimum, median, maximum, mean and standard deviation (SD) values for the amounts of specific types of feeds (kg DM per 100 kg body weight and day), total daily feed intake and total intake of feed components (grams per 100 kg of body weight per day unless otherwise stated in the table) for farm-matched pairs of horses with (case, *n* = 50) and without (control, *n* = 50) free faecal liquid. Not all horses were fed all listed feed types.

Variable	Case					Control					
	Minimum	Median	Maximum	Mean	SD	Minimum	Median	Maximum	Mean	SD	*p*-Value
Amount of specific type of feed ^1^											
Wrapped forage	0.4	1.4	2.5	1.4	0.44	0.4	1.4	3.5	1.5	0.61	0.97
Hay	0	0	0.6	0	0.11	0	0	1.3	0	0.19	NA
Straw	0	0	1.1	0.1	0.18	0	0	0.8	0.2	0.19	<0.0001
Lucerne	0	0	0.3	0.03	0.07	0	0	0.4	0.08	0.05	0.05
Concentrates	0	0.1	1.5	0.2	0.3	0	0.1	0.6	0.1	0.16	0.004
Mineral feeds	0	0	0	0.008	0.01	0	0.02	0.04	0.01	0.011	0.08
Total feed ration ^1,2^											
Total daily feed intake	0.4	1.6	3.7	1.7	0.61	0.4	1.6	3.6	1.9	0.62	0.09
Proportion of concentrate, % of diet	0.5	6.9	42.7	9.7	8.22	0.7	7.7	47.9	9.1	8.16	<0.0001
Total intake of feed component											
Metabolisable energy for horses, MJ	3	15	67	16	8	3	15	42	16	6.9	0.57
Digestible crude protein	3	81	337	89	50.6	3	81	285	95	49.4	0.007
Neutral detergent fibre	257	929	3459	1005	524.7	181	929	3237	1105	522.9	<0.0001
Starch	0	2	106	19	28.9	0	2	102	17	26	0.004
Crude fat	0	1	37	5	9	0	1	28	5	6.8	0.28
Water-soluble carbohydrates	20	172	367	177	80	26	152	359	167	74.5	0.02
Calcium (Ca)	1	8	47	10	6.4	2	8	34	10	6.3	0.47
Phosphorus (P)	1	4	15	5	2.8	1	4	16	5	2.7	0.85
Magnesium (Mg)	1	3	11	3	2.1	1	3	13	3	2.2	0.63
Sodium (Na)	0	1	19	2	2.6	0	1	22	2	3.2	0.92
Potassium (K)	5	26	126	29	18.7	5	26	151	27	16.7	0.95

^1^ Horses having ad libitum access to any feed were not included in these calculations. ^2^ On dry matter basis. NA, not applicable.

**Table 3 animals-11-02552-t003:** Distribution of feeding strategies of farm-matched pairs of horses with (case, *n* = 50) and without (control, *n* = 50) free faecal liquid.

Variables	Case, *n* (%)	Control, *n* (%)	*p*-Value
Number of feedings of forage per day			0.87
1 time	1 (2)	0 (0)	
2 times	15 (30)	13 (26)	
3 times	14 (28)	13 (26)	
4 times	14 (28)	14 (28)	
>4 times	6 (12)	8 (16)	
*Ad libitum* access	0 (0)	2 (4)	
Maximum time between two feedings of roughage			0.45
<4 h	7 (14)	9 (18)	
4–8 h	12 (24)	13 (26)	
>8 h	25 (50)	28 (56)	
Free access	0 (0)	2 (4)	
Feeding strategy for roughage in paddock			0.76
Forage not fed in the paddock	4 (8)	3 (6)	
On the ground	16 (32)	18 (36)	
In a feeding rack/tub or similar	23 (46)	23 (46)	
Combination of ground and feeding rack	3 (6)	1 (2)	
Other (in a hay-net, from a bale, in an automatic feeder)	4 (8)	4 (8)	
Number of concentrate feedings per day			0.43
Not fed concentrate	14 (28)	14 (28)	
1 time	18 (38)	21 (42)	
2 times	12 (24)	9 (18)	
3 times	5 (10)	4 (8)	
4 times	1 (2)	2 (4)	
>4 times	0 (0)	0 (0)	
Type of water source in stable/loose housing system			
Frostless automatic waterer	18 (32)	20 (40)	0.76
Automatic waterer	8 (16)	7 (14)	
Tub	2 (4)	3 (6)	
Bucket	13 (26)	10 (20)	
Natural water source	3 (6)	2 (4)	
Combination of bucket and automatic waterer	6 (12)	8 (16)	
Type of water source in paddock during winter			0.25
Frostless automatic waterer	12 (24)	13 (26)	
Frostless tub	14 (28)	16 (32)	
Automatic waterer	1 (2)	3 (6)	
Tub	11 (22)	10 (20)	
Bucket	2 (4)	4 (8)	
Natural water source	2 (4)	2 (4)	
Combination of bucket and automatic waterer	8 (16)	2 (4)	

**Table 4 animals-11-02552-t004:** Changes in the appearance of free faecal liquid in the horses in the study (*n* = 50) with diet changes, as reported by respondents. “Less loose” refers to reduced amount of liquid phase in faeces compared to before the feed change. Not all respondents had tried all listed changes, and some had tried more than one change.

Changes in Faecal Appearance	Case Horses, *n* (%)
Less loose when changing from wrapped forage to hay	9 (18)
Less loose when changing from wrapped forage to pasture	12 (24)
Less loose when changing to another batch of wrapped forage	4 (8)
No change in faecal appearance with any change in feeding	3 (6)
More loose in association with changing feeds	4 (8)
Less loose when changing from primary to regrowth harvest ^1^	17 (34)
Less loose when using feed additives ^2^	13 (26)
Have not tried any change in feeding	0 (0)

^1^ Wrapped forages, regrowth harvest defined as 2nd, 3rd or 4th harvest. ^2^ Feed additives reported included different types of commercial pro- and prebiotics, psyllium seed, thiamine and linseed.

**Table 5 animals-11-02552-t005:** Distribution of management strategies of farm-matched pairs of horses with (case, *n* = 50) and without (control, *n* = 50) free faecal liquid.

Variables	Case, *n* (%)	Control, *n* (%)	*p*-Value
Housing system			0.81
Individual box at night, in paddock during daytime	32 (64)	31 (62)	
Loose housing system	18 (36)	19 (38)	
Bedding			0.72
Straw	14 (28)	15 (30)	
Shavings	6 (12)	8 (16)	
Sawdust	6 (12)	5 (10)	
Peat	3 (6)	2 (4)	
Combination of shavings and peat	3 (6)	4 (8)	
Rubber mat	3 (6)	2 (4)	
Combination of shavings and straw	12 (24)	11 (22)	
Other (Raw sawdust, straw pellets)	3 (6)	3 (6)	
Access to salt lick in loose housing system/stable			1.00
Yes	46 (92)	45 (90)	
No	4 (8)	5 (10)	
Time spent per day in paddock during winter			0.51
<8 h	14 (28)	19 (32)	
8–12 h	17 (34)	15 (30)	
>12 h	19 (38)	19 (38)	
Paddock type (winter)			0.67
Grass (old grass during winter)	15 (30)	17 (34)	
Sand/Gravel	6 (12)	6 (12)	
Soil	16 (32)	15 (30)	
Forest	13 (27)	12 (24)	
Annual time spent on pasture			0.33
<4 weeks	2 (4)	2 (4)	
4–8 weeks	11 (22)	7 (14)	
8–12 weeks	8 (16)	9 (18)	
>12 weeks	23 (46)	23 (46)	
Not on pasture	6 (12)	9 (18)	
Type of pasture			0.68
Pasture on arable land	9 (18)	10 (20)	
Natural or semi-natural pasture	19 (38)	17 (34)	
Forest pasture	1 (2)	1 (2)	
No pasture	5 (10)	9 (18)	
Other (combination of different pasture types)	16 (32)	13 (26)	
Type of water source on pasture			0.43
Frostless automatic waterer	2 (4)	0 (0)	
Frostless tub	3 (6)	3 (6)	
Automatic waterer	3 (6)	1 (2)	
Tub	24 (48)	25 (50)	
Bucket	5 (10)	5 (10)	
Natural water source	3 (6)	6 (12)	
Combination of automatic waterer/bucket or automatic waterer/tub	10 (20)	10 (20)	
Access to salt lick while on pasture			0.30
Yes	43 (86)	40 (80)	
No	7 (14)	10 (20)	
Anthelmintic routines			0.37
Regularly dewormed ≥ 1 time per year	10 (20)	14 (28)	
Dewormed due to high ^1^ egg counts ≥ 1 time per year	27 (54)	25 (50)	
Dewormed due to high ^1^ egg counts < 1 time per year	8 (16)	6 (12)	
Dewormed if considered necessary	3 (6)	3 (6)	
Not dewormed	2 (4)	2 (4)	
Time from last deworming			0.70
Not dewormed	2 (4)	2 (4)	
0–3 months ago	22 (44)	18 (36)	
3–6 months ago	12 (22)	13 (24)	
6–12 months ago	2 (4)	7 (14)	
>1 year ago	12 (24)	10 (20)	

^1^ According to national guidelines (www.sva.se (accessed on 17 February 2017)).

## Data Availability

The authors confirm that all source data will be deposited into an institutional data repository and made available upon request to the corresponding author.

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
