# Peer review of "Feeding and Management of Horses with and without Free Faecal Liquid: A Case–Control Study"

_animals, 2021, doi:10.3390/ani11092552_

Round 1
Reviewer 1 Report
General comment: The manuscript “Feeding and management of horses with and without free faecal liquid- a case-control study” is interesting but not really original. The study is mainly an observational one, and the contents addressed in this study could be worthy of investigation mainly from the practical point of view. The planning adopted to fulfill the investigation presents some limitations that are correctly and clearly presented by the Authors, although the study was well organized and made as presented by the exhaustive description. The findings presented are enough clear. The results proposed by the authors are extensively described and substantial, confirming similar results yet obtained. The single points are well discussed and compared to a limited not really up-to-date literature on the argument.
Title: appropriate
Abstract: suitable
Introduction: This section is concise, and includes the specific literature references. An hypothesis could be added to better explain the methodology adopted to fulfill the aim of the research.
Material and methods: The study was well organized and made as presented by the exhaustive description. The methodologies adopted are the current ones. Nevertheless, the planning adopted to fulfill the investigation presents some limitations that are correctly and clearly presented by the Authors.
Results: The results obtained by the authors are extensively described and substantially commented in the light of the aim of the survey adopted.
Discussion: The comments reported in discussion, although concise, are pertinent to the results achieved, but they do not give any final achievement.
References: are appropriate, although limited and not completely up-to-date.
Figures and Tables: are explicative.
Decision: Although the current manuscript presents a limited originality, it is acceptable for publication.
Reviewer 2 Report
This is an interesting paper and you have compiled a significant amount of information. I would have liked to have had access to the table with the horse data which needs to be included.
